# Evaluation of Blood C Reactive Protein (CRP) and Neutrophil-to-Lymphocyte Ratio (NLR) Utility in Canine Epilepsy

**DOI:** 10.3390/vetsci11090408

**Published:** 2024-09-04

**Authors:** Andreea Despa, Mihai Musteata, Gheorghe Solcan

**Affiliations:** 1Neurology Service, Faculty of Veterinary Medicine, Ion Ionescu de la Brad Iași University of Life Sciences (IULS), 700489 Iași, Romania; andreea.despa@iuls.ro; 2Internal Medicine Clinic, Faculty of Veterinary Medicine, Ion Ionescu de la Brad Iași University of Life Sciences (IULS), 700489 Iași, Romania

**Keywords:** C reactive protein, neutrophil-to-lymphocyte ratio, epilepsy, neuroinflammation, inflammation markers, canine

## Abstract

**Simple Summary:**

Canine epilepsy is one of the most common chronic neurological diseases. In the last few decades, research has focused on revealing the role of neuroinflammation in the development of seizure activity and the biomarkers accompanying this process. The purpose of this study was to investigate if the blood C reactive protein concentration and the neutrophil-to-lymphocyte ratio in epileptic dogs might be useful first-line tools for preliminary discrimination between the main etiologies of canine epilepsy. The neutrophil-to-lymphocyte ratio was found to be increased in all epileptic patients, regardless of the etiology or clinical presentation. In contrast, the C reactive protein concentration was abnormal only in dogs with cluster seizures due to idiopathic epilepsy and structural epilepsy. These results could be useful for general practitioners in adapting their therapeutic approach.

**Abstract:**

Background: The role of neuroinflammation in epileptogenesis has been previously explored, and several biomarkers have been identified as being relevant in assessing the intensity of the inflammatory process. In human medicine, an increased C reactive protein (CRP) blood concentration and/or neutrophil-to-lymphocyte ratio (NLR) is considered a constant finding of epileptic activity. In veterinary medicine, only a few studies have been published regarding both of these topics. Hypothesis/objectives: Our aim was to assess the C reactive protein blood concentration and the neutrophil-to-lymphocyte ratio in epileptic dogs, regardless of etiology. Method: This retrospective study was based on changes in routine blood parameters in 59 dogs with epileptic activity. Results: An increased C reactive protein concentration was observed mostly in the dogs affected by structural epilepsy, and all epileptic dogs displayed abnormal neutrophil-to-lymphocyte values. Conclusions: Based on the authors’ knowledge, this is the first report regarding the NLR in epileptic dogs. Both the CRP concentration and the NLR might be considered feasible non-specific markers of the neuroinflamation involved in epileptogenesis and might be used in the diagnosis of and therapeutic approach to cluster seizures in dogs with idiopathic epilepsy and in patients with structural epilepsy. Dogs diagnosed with IEis and high CRP concentrations and NLRs may be subject to non-documented cluster seizures. Both CRP and the NLR have limited diagnostic value in dogs with reactive seizures.

## 1. Introduction

Epilepsy is one of the most common chronic neurologic disorders in dogs, with an estimated prevalence of 0.6–0.75% in the entire dog population [1,2]. Both in humans and in animal models, inflammation has been demonstrated to be a part of its pathogenesis, with a significant role in the initiation and development of seizures [3]. Hence, epileptogenesis is correlated with neuronal damage, gliosis, and inflammation of the central nervous system and systemic circulation associated with blood–brain barrier (BBB) breakdown [4,5]. These inflammatory and cytotoxic neural processes contribute to the production of pro-inflammatory cytokines (IL-1β, IL-2, IL-6, and TNF-α), which subsequently lead to increased levels of acute-phase proteins (APPs), as part of an early non-specific immune response [6,7].

APPs vary in response magnitude and type across animal species, with the liver being the primary site of their production. In dogs, CRP and serum amyloid A (SAA) are considered to be major APPs, while fibrinogen and haptoglobin are classified as moderate APPs. However, only a few studies have analyzed the role of different APPs in various neurological diseases. For example, in one study investigating the diagnostic utility of D-dimer and CRP concentration both in blood and cerebrospinal fluid (CSF) in neurological diseases, the authors reported high CSF D-dimer and CRP concentrations only in dogs with inflammatory conditions. In addition, in intracranial neoplasia, the CSF D-dimer concentration was significantly higher when compared to a control group [8]. Other diagnostic markers, such as genetic markers, SAA, cytokines such as interleukin-17 and CC-motif ligand 19, endocannabinoid receptors, and heat shock protein 70, have been proposed in some inflammatory neurological conditions [9,10], but they are not currently used in general practice due to their lack of specificity [11]. CRP in particular exhibits the most significant increase in dogs, making it a highly valuable biomarker for the diagnosis and prognosis of various diseases in this species [12]. To the best of our knowledge, there are no reports regarding the influence of individual factors such as age and sex on the CRP concentration in dogs, and with the exception of Miniature Schnauzers, which tend to display a slightly higher median CRP concentration [13], interbreed differences have not been identified.

The CRP concentration has been investigated in dogs diagnosed with idiopathic (IE) or structural epilepsy (SE). Even though the CRP values were higher in dogs with cluster seizures or SE, studies have not identified a statistical difference compared to IE dogs in terms of the frequency of seizures [14,15]. Beside CRP, the neutrophil-to-lymphocyte ratio (NLR) is another potential marker of systemic inflammation. In humans, when compared with a healthy group, the overall NLR was found to be higher in epileptic patients, especially in the postictal phase [3,16]. These results suggest that an increased excitability of neuronal cells might be connected with neutrophil-mediated systemic inflammation [3]. Another study investigating the changes found in the routine blood parameters of human patients with generalized tonic-clonic seizures reinforced previous findings and underlined a relationship between the platelet-to-lymphocyte ratio and neutrophil-mediated inflammation, concluding that an increase in the NLR by one unit corresponds to 1.95-fold increase in the risk of seizures [16]. Therefore, by determining the NLR, the body’s inflammatory response to the host immune response could be assessed [17]. Despite the breadth of research conducted in human medicine, in animals, only a few studies have investigated the NLR’s utility as a diagnostic and prognostic marker in neoplastic [17,18,19,20,21,22], cardiological [23], pulmonary [24], gastro-intestinal [25,26,27], septic, and inflammatory diseases [28,29,30], and only one study has evaluated the NLR’s diagnostic value in dogs with meningoencephalithis of unknown origin (MUE) [31]. However, that study’s purpose was to investigate the NLRs in relationship to the primary etiology but not the clinical symptomatology of seizures. Furthermore, as of present, there is no study investigating the diagnostic utility of the NLR in dogs with epilepsy. 

Therefore, our aim was to evaluate both the CRP activity and NLR in epileptic dogs with respect to the etiology and type of seizure and to uncover if the NLR might be used as a diagnostic marker in differentiating the main types of epilepsy. 

## 2. Materials and Methods

### 2.1. Case Selection and Grouping

Clinical records of dogs presented to IULS Veterinary Teaching Hospital to investigate their epileptic activity from September 2021 to January 2023 were reviewed. Ethical approval for the study was obtained from the Ethics Committee of the Faculty of Veterinary Medicine, University of Life Sciences “Ion Ionescu de la Brad” from Iași (no. 454/14.03.2022). The included dogs had a history of at least two unprovoked epileptic seizures occurring at least 24 h apart and had underwent complete physical and neurological examination, urinalysis, supplementary thoracic radiography, abdominal ultrasound, and measurements of their complete blood cell count, serum biochemical profile, and blood pressure; therefore, all dogs met the first confidence level according to International Veterinary Epilepsy Task Force (IVETF) consensus [32]. Although recommended, the second and the third diagnostic levels of confidence [32] were not mandatory.

SE group was represented by dogs in which an underlying cerebral disease was suspected based on their neurological examination and clinical course and supported by paraclinical findings, including CBC measures, CSF analysis, infectious disease screening (serology or PCR), electroencephalography, and/or advanced imaging (CT/MRI). This category included vascular, inflammatory/infectious, traumatic, anomalous, and neoplastic CNS diseases [1,33]. Reactive epilepsy (RE) group included patients that developed seizures due to metabolic disorders. Cases with no evidence of a cerebral lesion and cases with no metabolic abnomalities were classified as dogs with idiopatic epilepsy (IE). 

Cluster epilepticus is defined as the presence of more than two self-limiting seizures over a period of 24 h [34]; therefore, its clinical features resemble a typical epileptic attack. Based on this, the IE group was further divided according to the frequency of the seizures as follows: IE with isolated seizures (IEis) and IE with cluster seizures (IEcs). Dogs with status epilepticus, a neuroemergency characterized by self-sustained, prolonged epileptic activity that involves a certain degree of neuroinflammation [8,9], were excluded from further analysis in both groups.

Dogs with incomplete anamnesis and/or medical records and dogs undergoing therapy with anti-inflammatory or immunosuppressive medication were excluded. 

Based on these findings, all dogs had a neuroanatomical localisation in the forebrain.

The control group was represented by 8 client-owned healthy dogs that presented for blood donation.

### 2.2. NLR Measurement

Blood samples were collected from the jugular or cephalic vein and a complete blood count (CBC) was performed using VetScan HM5 Hematology Analyzer (Zoetis, Union City, CA, USA), at first medical visit. For all dogs, the neutrophil and lymphocyte counts were extracted from CBC and calculated as follows: NLR = Neutrophil count/Lymphocyte count. The N:L cut-off value was determined by the values obtained from the control group.

### 2.3. Serum CRP Measurment

The peak concentration of serum CRP is estimated at 24–48 h in the majority of the cases, therefore the maximum period interval for performing the test was 48 h from the last seizure [35,36,37]. A commercially available assay Vcheck Canine CRP (Bionote, Hwaseong-si, Republic of Korea) validated for dogs was used in order to measure the CRP concentration [38]. The manufacturer reference intervals are as follows: <10 mg/L—normal; 10–20 mg/L—normal; 20–30 mg/L—equivocal; and >30 mg/L—abnormal. For statistical purposes, the values under 10 mg/L were replaced with 10.

### 2.4. Data Collection and Analysis

Collected data for all dogs included age, sex, breed, description of seizures, whether they received antiepileptic drugs (AEDs), white blood cell (WBC) count, neutrophil count, lymphocyte count, and CRP concentrations. After testing for normality (Shapiro–Wilk test), the Mann–Whitney U test was used for comparisons between the groups, and the Kruskal–Wallis test was used for comparisons between three or more groups. The obtained values are presented as means with standard error. A *p*-value lower than 0.05 was considered statistically significant. All analyses were performed with Statistical Package for the Social Science (SPSS) 26 for Windows from IBM.

## 3. Results

### 3.1. Study Population

A total of 59 dogs with epileptic activity met the inclusion criteria and were categorised as IE (n = 29), SE (n = 20), or RE (n =10). The IEis group was represented by 23 dogs: 16 males and 7 females, with a median age of 4.6 years (range: 1–14 years) of different breeds (four mixed breed, four Maltese Bichons, three Pugs, two Labrador Retrievers, two Beagles, and one of each of the following: Akita Inu, Amstaff, French Bulldog, Mioritic Sheperd, Caniche, Spanish Cocker, Yorkshire Terrier, and Greyhound). The IEcs group consisted of six dogs: five males and one female, with a median age of 2.5 years (range: 2–3 years), of different breeds (three Cane Corso, one Caucasian Sheperd, one Maltese Bichon, and one mixed breed). The SE group consisted of 20 dogs: 9 males and 11 females, with a median age of 5.5 years (range: 1–14 years), of different breeds (six mixed breeds, two Beagles, four French Bulldogs, one German Sheperd, two Maltese Bichons, two Pugs, and one of each of the following: Yorkshire Terrier, Kangal Sheperd, and Pomeranian) (see Appendix A, Epidemiological, clinical and diagnostic features of dogs from structural epilepsy group). The RE group included ten dogs, nine females and one male, diagnosed with hepatic encephalopathy (n = 3), hypothyroidism (n = 2), pyometra (n = 1), and cardiopulmonary failure (n = 4). 

### 3.2. Serum CRP Concentration

The serum CRP concentration had a normal value (under 20 mg/L) in 22/23 of the dogs from the IEis group; therefore, only one dog had an equivocal value (21.5 mg/L). Meanwhile, for the IEcs group, four of the six dogs had normal values and two of the six had abnormal values (31.4 mg/L and 57.9 mg/L, respectively). Half of the dogs (10/20) with SE had modified CRP concentrations: five had abnormal values (above 30 mg/L) and five had equivocal values (20–30 mg/L). The rest of the 10 dogs had CRP values in normal ranges (one value between 10 and 20 mg/L and nine values of 10 mg/L). Only two of the ten dogs from the RE group obtained an abnormal CRP value and one of the ten had an equivocal value, while seven out of the ten dogs had normal CRP blood concentrations (Table 1).

Therefore, the CRP concentration of the SE patients was significantly increased in comparison to that of the IE group (mean ± standard error: 46.83 ± 13.56 mg/L vs. 12.42 ± 0.89 mg/L; *p* = 0.008). When the CRP values were compared between the RE and SE groups, the CRP concentration was found to be increased in the SE dogs, but no statistically significant difference was observed (46.83 ± 13.56 mg/L vs. 28.17 ± 13.76 mg/L, *p* = 0.57). Finally, the RE group had higher CRP values when compared to those of the IE group (*p* = 0.265) (Figure 1). When the IE subgroups were compared, the EIcs group had a statistically higher CRP value in comparison with EIis (17.41 ± 3.30 mg/L and 11.11 ± 0.52 mg/L, respectively; *p* = 0.0022) (Figure 2).

### 3.3. Comparison of the NLR between Healthy Dogs and Epileptic Dogs

The median NLR cutoff was considered to be 3.84 ± 0.36, based on the values obtained from the control group; the median values of the healthy group were 9.66 ± 0.50 × 10^9^/L for neutrophils and 2.74 ± 0.38 × 10^9^/L for lymphocytes. 

Of all the groups with epileptic activity, the SE group’s neutrophil count was statistically increased when it was compared with the values of healthy dogs (mean ± standard error: 13.09 ± 0.78 × 10^9^/L vs. 9.66 ± 0.50 × 10^9^/L; *p* = 0.015). For the IE and RE groups (10.26 ± 0.78 × 10^9^/L and 13.60 ± 4.97 × 10^9^/L, respectively), the values were higher than those of the control group, but no statistical significance was observed (*p* = 0.941 and *p* = 0.829, respectively). When we compared the values between the dogs with seizure activity, the SE group’s neutrophil count was significantly increased in comparison with the neutrophil counts from the IE (*p* = 0.023) and RE (*p* = 0.029) groups. However, we identified a higher neutrophil count in the IE group than in the RE group, but there was no statistically significant difference (*p* = 0.748). Moreover, the values were lower only in the dogs from the IEis group and higher in the IEcs group (8.76 ± 0.66 × 10^9^/L vs. 16.02 ± 0.94 × 10^9^/L; *p* = 0.001).

The lowest lymphocyte values among all groups were registered in the IE and SE (1.79 ± 0.13 × 10^9^/L, 1.89 ± 0.21 × 10^9^/L) groups in comparison with the healthy group (2.74 ± 0.38 x 10^9^/L; *p* = 0.039, and *p* = 0.037, respectively). We did not identify a significant increase when comparing the RE group (2.91 ± 1.14 × 10^9^/L) with the healthy group (*p* = 0.122). Furthermore, no statistical differences were observed when we compared the IE subgroups. Likewise, the IEcs group displayed an increase in comparison with the IEis group, but without a significant difference (2.10 ± 0.29 × 10^9^/L vs. 1.71 ± 0.14 × 10^9^/L; *p* = 0.196) (Table 2).

The entire group with seizure activity had a significantly greater NLR compared to the healthy group (mean ± standard error: 3.84 ± 0.36; *p* = 0.005) (Figure 3). When compared to the control group, both the IE and SE groups (6.49 ± 0.69; 8.19 ± 0.86) displayed a statistically increased NLR (*p* = 0.001 and *p* = 0.012, respectively), but not the RE dogs (5.66 ± 1.27; *p* = 0.274). No statistical differences were observed when the following groups were compared: IE with SE (*p* = 0.087) and RE (*p* = 0.85), even if the mean NLR of the SE group was higher than that of the RE group. In addition, the mean NLR of the IEcs group was increased when compared with the IEis group, but without a statistical difference (8.69 ± 1.74 vs. 5.92 ± 0.71; *p* = 0.085).

## 4. Discussion

In this paper, we have investigated both the CRP activity and NLR in dogs with epilepsy with respect to etiology and symptomatology. Furthermore, we analyzed if the NLR might be considered as a feasible and cost-effective marker for diagnosing the main types of canine epilepsy.

Our results show that a third of the dogs diagnosed with IEcs had an abnormal CRP blood concentration when compared with the IEis group, supporting the results from two other studies [14,15]. Similar to the healthy dog group, all dogs with IE had a lower median CRP value in comparison with that of the SE group. Furthermore, we found that a half of the dogs diagnosed with either neoplasia or an inflammatory intracranial disease included in the SE group had an abnormal basal blood CRP value, similar to a previous report in which 62% of dogs with structural epilepsy had a higher blood CRP concentration [15]. Cavalerie, R. et al. (2024) showed that CRP has a limited diagnostic value in dogs with meningoencephalitis of unknown origin, in which only 30% of dogs had an increase in blood CRP concentration [39]. However, our aim was not to subdivide the SE group by pathologies, but to compare it with the findings from other epileptic groups. Hence, the high CRP concentration identified in our study might be a consequence of the heterogeneity of the SE group (e.g., differences in the underlying conditions or stages of disease). Moreover, our study did not include dogs with status epilepticus but only those with self-limiting seizure activity (IEis and IEcs). Previous research studies suggest that status epilepticus might increase the CRP’s concentration as a result of seizure-induced neuroinflammation regardless of etiology [14,15]. Therefore, we cannot exclude a higher CRP activity in dogs with status epilepticus, similar to that of dogs with SE. 

Lastly, the majority of the RE group had normal CRP values, with only two dogs having abnormal values due to hepatic encephalopathy secondary to liver neoplasm or a concomitant infectious disease. For these patients, the high CRP values were most likely related to the primary liver or intestinal pathologies and not the seizure activity [35,40,41]. However, due to both the small number of dogs with RE included in our study and the high variability in CRP, it was difficult to assess the dynamic of the CRP concentration in this epileptic group.

Despite the fact that we identified high CRP values in both the SE and IEcs groups, we failed to recognize CRP as a reliable indicator of the neuroinflammatory processes involved in epileptogenesis. Therefore, we investigated if the NLR, another inflammatory biomarker, can be used as an additional diagnostic tool of canine epilepsy.

Using dogs as a natural animal model for human epilepsy [42], especially for patients with drug-resistant epilepsy (encountered both in humans and canines), is a new topic that involves translating the knowledge gained from studies on people and rodents to dogs and vice versa [43]. To this point, inflammation has been established to play an important role in the development of seizure activity in both human and animal models [3,4], but the research regarding the biomarkers which reflect this process is still in early stages [44,45,46]. Therefore, understanding epileptogenesis and the role of inflammation in generating seizures might be helpful in developing new therapeutic strategies. Previous research has shown that the infiltration of inflammatory cells into the hippocampus, the breakdown of the blood–brain barrier (BBB), and increasing levels of leukocytes predispose patients to neurodegeneration [47,48,49]. Furthermore, the potential of therapies such as anti-inflammatory drugs is being investigated in order to establish if they could modulate the upregulation of the BBB by COX-2 inhibition [50,51].

In our study, all epileptic groups, regardless of etiology, had an increased NLR (*p* = 0.005). The analysis of the NLR is preferable to the lymphocyte or neutrophil count alone because it offers a more nuanced and comprehensive assessment of the body’s immune response [17], which might be helpful in the diagnosis, prognosis, and monitoring of the disease. Indeed, our results showed no significant differences in the lymphocyte count when both the groups and subgroups were compared. Therefore, the increased NLR was based only on the neutrophil count. Thus, we suggest that the lymphocyte count may not be a reliable indicator of epileptic activity and should be combined with other markers to obtain a better diagnostic accuracy. As we expected, after investigating the CRP values, similar to reports in humans, the greatest statistical difference was found between the NLRs of the SE group and the control group. The majority of the dogs included in this category were diagnosed with either neoplasia or an inflammatory intracranial disease, mainly based on the clinical course and the findings from the cytological examination of the CSF or the MRI/CT study of the brain. As it has been shown in previous reports, immunosuppression and remodelation generate a tumour microenvironment which lead to neuroinflammation [52,53,54]. Furthermore, it has been suggested that neutrophils are involved in the early stages of its pathogenesis, accelerating the infiltration of the inflammatory cells and therefore generating lesions [55]. Thus far, the contribution of the neutrophils in the neuronal regulation of autoimmunity remains unclear [56,57]. As a result of the similarities of inflammatory brain diseases in humans and dogs [43], we presume that the NLRs are related to neuronal inflammation.

As mentioned by Charalambous et al. [34], inflammation plays an important role in the maintenance and exacerbation of ongoing seizure activity from both cluster and status epilepticus [34,42,58]. Therefore, we believe that the BBB breakdown related to these neurological emergencies is the main reason for the high NLR blood values we found in the IEcs group. Surprisingly, the NLR was also increased in the EIis group, when compared with the healthy dogs. This finding might suggest that although an epileptic seizure is defined as a self-limitating excessive activity, the neutrophils and the microglial activation might be part of the ictogenesis. However, all the CBCs were performed in the postictal phase and the time frame could influence our findings, considering the catecholamine release, the leukocyte migration from the blood to brain, and the neutrophil adhesion in brain capillaries, all which accompany an epileptic event [48,59,60]. Future research studies are needed in order to identify if the inflammatory biomarkers are at the origin of an epileptic seizure rather than a consequence of one. Thus, our results suggest that the NLR might be an useful first-line tool, for example, for general practitioners in order to preliminarily discriminate between the main etiologies of canine epilepsy and therefore adapt their therapeutic approach. Furthermore, due the fact that in our dogs both the NLR and CRP were significantly increased in cluster patients (*p* = 0.001), we recommend that dogs that present with a history of isolated seizures and elevated NLR and CRP levels to be managed as cluster seizures patients and to be hospitalized for monitoring for at least 48 h. This approach is crucial because, in veterinary medicine, one of the primary limitations is that owners may not witness or accurately report the true number of seizures their pet has experienced within the last 24 h. Consequently, these dogs may be at risk of experiencing a neurological emergency, such as non-documented cluster seizures.

Our study had several limitations. The most important one is represented by the small number of dogs in each group included in this study; further studies based on larger cohorts included in multicentric studies might be useful in highlighting the statistical significance of both the CRP concentration and NLRs according to seizure etiology. Second, the SE group was not subdivided according to the primary pathology; therefore, different results might be expected, considering the level of neuroinflammation of various structural pathologies. Third, all the CBC tests occured in the postictal phase, without assessing their kinetics over the hospitalization period. Similar to human medicine studies, we appreciate that the NLR might decrease and reach a physiological value in the interictal phase [16]. Lastly, we only investigated dogs with self-limiting seizures (IEis and IEcs). Our results are not necessarily applicable in dogs with status epilepticus regardless of etiology.

## 5. Conclusions

In conclusion, in the absence of other concurrent inflammatory diseases, the blood CRP concentration and NLR might be considered as potential non-specific markers and used in the diagnostic approach of dogs with structural epilepsy or cluster seizures in dogs with idiopathic epilepsy. Dogs diagnosed with IEis and high CRP concentrations and NLRs may be subject to non-documented cluster seizures. Both measurements have limited diagnostic value in dogs with reactive seizures.

## Figures and Tables

**Figure 1 vetsci-11-00408-f001:**
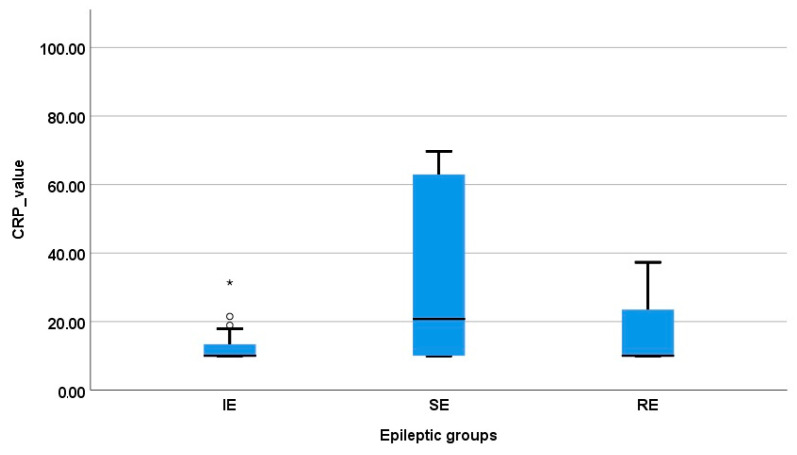
Distribution of CRP concentration in the epileptic groups. Three dogs (*, ^o^ marks) previously diagnosed with IE had a CRP value out of the group mean ± standard error but no other pathology was identified in their cases.

**Figure 2 vetsci-11-00408-f002:**
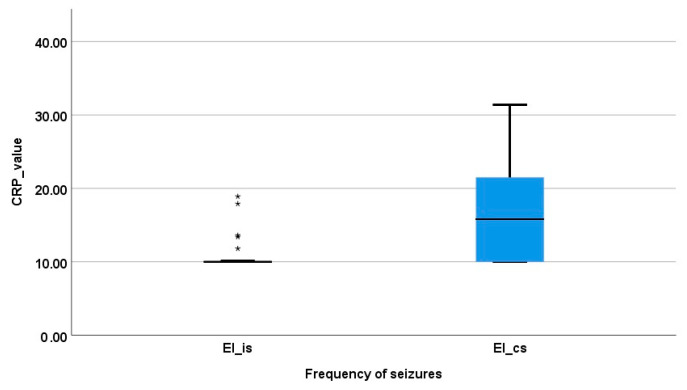
Distribution of CRP concentration in the idiopathic epilepsy subgroups. Seven dogs (* marks) previously diagnosed with IE with isolated seizures had a CRP value out of the subgroup mean ± standard error but no other pathology was identified in their cases.

**Figure 3 vetsci-11-00408-f003:**
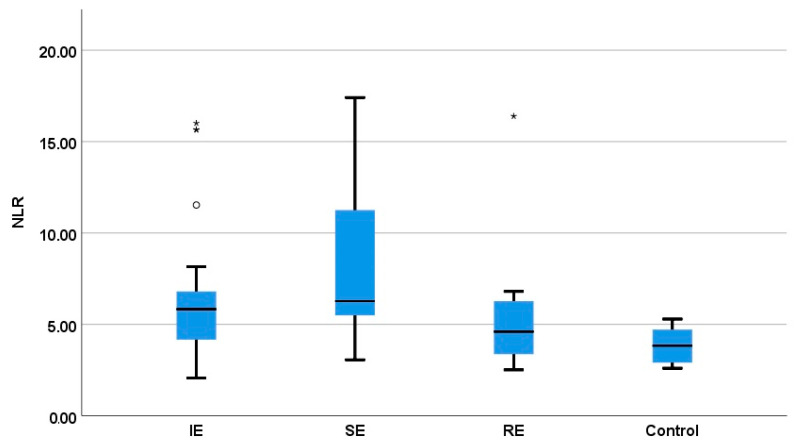
Comparison of NLRs in correlation with the etiology of epileptic seizures. Four IE dogs and one with RE (*, ^o^ marks) had a NLR value out of the coresponding group mean ± standard error but no other pathology was identified in their cases.

**Table 1 vetsci-11-00408-t001:** CRP and NLR percentages in all epileptic groups.

Percentage of Dogs with:				
	CRP Value		NLR	
	Normal	Equivocal/Abnormal	Normal	Abnormal
IEis	95.70%	4.30%	30%	70%
IEcs	66.66%	33.33%	-	100%
SE	50%	50%	5%	95%
RE	60%	40%	30%	70%

**Table 2 vetsci-11-00408-t002:** NLR value in dogs with idiopathic epilepsy.

	IE Isolated Seizures (n = 23)	IE Cluster Epilepticus (n = 6)	*p* Value
Neutrophil count (× 10^9^/L), Median ± standard error (Range)	8.76 ± 0.66 (3.7–17.54)	16.02 ± 0.94 (13.18–19.36)	**0.001**
Lymphocyte count (× 10^9^/L), Median ± standard error (Range)	1.71± 0.14 (0.75–3.04)	2.10 ± 0.29 (1.21–3.04)	0.196
NLR, Median ± standard error (Range)	5.92 ± 0.71 (2.06–15.64)	8.69 ± 1.74 (5.71–16.01)	0.085

## Data Availability

The original contributions presented in the study are included in the article, further inquiries can be directed to the corresponding authors.

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
