# Peer review of "Evaluation of Blood C Reactive Protein (CRP) and Neutrophil-to-Lymphocyte Ratio (NLR) Utility in Canine Epilepsy"

_vetsci, 2024, doi:10.3390/vetsci11090408_

Round 1

Reviewer 1 Report

Comments and Suggestions for Authors

Interesting work, but I think that writing "Evaluation of Blood Protein C reactive (CRP) and Neutrophil-to-Lymphocyte Ratio (NLR) as a diagnostic tool in canine epilepsy" is a bit excessive. As the authors point out, a limitation of the study is the small number of dogs. I would like the authors to provide a better description of the dogs in the study (history), and to list the primary pathologies of the SE group. The lack of this data could influence the interpretation of the results.

Author Response

We would first like to thank you for the consideration, suggestions and kind remarks regarding the submitted manuscript. We resubmit now the revised form of the manuscript. Bellow you may find our point by point answer to all the comments and suggestions you had.

Reviewer 1:

Comment 1

Interesting work, but I think that writing "Evaluation of Blood Protein C reactive (CRP) and Neutrophil-to-Lymphocyte Ratio (NLR) as a diagnostic tool in canine epilepsy" is a bit excessive

Response 1

Thank you for your appreciations and suggestions. In the actual form of the manuscript we change the title as follow:

“Evaluation of Blood C reactive protein (CRP) and Neutrophil-to-Lymphocyte Ratio (NLR) utility in canine epilepsy”

We hope that you might find the actual title more suitable.

Comment 2

As the authors point out, a limitation of the study is the small number of dogs. I would like the authors to provide a better description of the dogs in the study (history), and to list the primary pathologies of the SE group. The lack of this data could influence the interpretation of the results.

Response 2

Thank you for pointing this out. Despite being performed on a small sample size, our study revealed a couple of statistically significant differences between the groups, which are expected to be more nuanced while being assessed on larger cohorts.

In the limitation section, line 313-316, we modified the manuscript and mentioned that future studies based on larger sample sizes should be performed in order to validate our findings: “Our study had several limitations. The most important one is represented by the small number of dogs in each group included in this study; future studies based on larger cohorts included in multicentric studies might be useful in highlighting a statistical significance of both the CRP and NLR values according to the etiology of the seizures.”

The history is explained in inclusion criteria, respectively for dogs diagnosed with IE between line 84-85: “The included dogs had a history of at least two unprovoked epileptic seizures occurring at least 24 hours apart”, for SE dogs  between line 91-96: “SE group was represented by dogs in which an underlying cerebral disease was suspected based on the neurological examination and clinical course and supported by paraclinical findings, including CBC, CSF analysis, infectious disease screening (serology, PCR), electroencephalography, and/or advanced imaging (CT/MRI). This category included vascular, inflammatory/infectious, traumatic, anomalous and neoplastic CNS diseases”, respectively for RE group between lines 96-97: “Reactive epilepsy (RE) group included patients that developed seizures due to metabolic disorders” and lines 150-152: “RE group included 10 dogs, 9 females and only 1 male diagnosed with hepatic encephalopathy (n = 3), hypothyroidism (n = 2), pyometra (n = 1) and cardiopulmonary failure (n = 4)”. For a better description of the dogs in the study, we attached a supplementary file with the description of the SE patients.

Thank you for your work and help in improving the quality of the manuscript!

Reviewer 2 Report

Comments and Suggestions for Authors

Title of the manuscript: Evaluation of Blood C reactive protein (CRP) and Neutrophil-to-Lymphocyte Ratio (NLR) as a diagnostic tool in canine epilepsy

The manuscript investigates the potential use of CRP level and neutrophil-to-lymphocyte ratio as diagnostic tools in serious conditions such as canine epilepsy. The study includes a retrospective analysis of 59 different-breeds dogs with epileptic activity. Animals are categorized into idiopathic epilepsy (IE), structural epilepsy (SE), and reactive epilepsy (RE) groups. The authors aim to evaluate whether acute phase proteins (APPs) such as CRP and NLR can be used to differentiate between these types of epilepsy.

The study is important because it addresses a significant gap in veterinary neurology, specifically the need for reliable biomarkers to differentiate between types of epilepsy in dogs. Nowadays there is no such marker and also clinical evaluation is complicated and expensive. Thus the topic is very important for small animals’ clinicians.  

I feel that a comprehensive data collection may be recognized as beneficial. The inclusion of multiple groups of epileptic dogs together with healthy control groups strengthens the study’s findings and provides a comprehensive comparison. However, in specific comments, I added some suggestions about that.

I admire that the manuscript effectively incorporates recent publications, which significantly enhances the relevance and credibility of the study's conclusions. By using up-to-date references, the authors have ensured that their work is aligned with the latest advancements in the field, providing a current and well-supported perspective on the use of APPs as diagnostic tools in canine epilepsy detection. This approach strengthens the study's overall contribution to veterinary science and highlights its importance.

Unfortunately, there are some weaknesses in the work such as:

-          Discussion on Long-Term Implications:

While the study’s conclusions regarding the use of CRP and NLR as diagnostic tools are supported by the data, the discussion could benefit from further exploration of the long-term implications of using these markers in everyday clinical practice. Thus, it would be beneficial to expand the discussion to include potential long-term implications and limitations of using CRP and NLR as diagnostic tools in canine epilepsy. This could include considerations of how these markers might perform in a broader clinical setting or over extended periods.

-          Sample Size:

The study’s conclusions are based on a relatively small sample size, which may limit the generalizability of the findings. However, future studies with larger sample sizes are recommended to validate the findings and to ensure that the results apply to a broader population of epileptic dogs.

-          Lack of summary including limitation part

-          Lack of brief explanation about acute phase protein usage in other species and species-dependent differences in this matter.

Specific Comments:

Line 30-35: In the introduction, there is a mention of the importance of CRP and NLR as potential diagnostic tools. However, it would be helpful if the authors provided a brief comparison with other commonly used inflammatory biomarkers in veterinary medicine, such as fibrinogen or haptoglobin, to contextualize why CRP and NLR were chosen for this study. The Introduction will benefit from that.

Line 50: Please add some information about acute phase proteins (APPs). Why we are using motly the major APPs as diagnostic markers? The authors should clearly explain that acute-phase proteins are non-specific markers. In different species of animals, we use different acute-phase proteins because there are species-specific differences. In horses, the main acute-phase protein is serum amyloid A (SAA), the changes in concentration of which are useful in detecting non-specific systemic bacterial infections but are not particularly helpful in fitness monitoring (ex. anabolic index is more helpful in this matter). In cows, SAA is also the main acute-phase protein, as well as in cats. However, in dogs, as in humans, the main acute-phase protein is CRP. Please find accurate literature to built this paragraph.

Line 57-58: The authors mention that previous studies did not identify a statistical difference in CRP values between IE dogs by frequency of seizures. It would be helpful to discuss why this might be the case and how this finding aligns or conflicts with the current study's results.

Line 122-129: please mention that the device was validated for canine species. Including this information about the validation of the assay would strengthen the reliability of the results presented.

Line 150-155: The diversity of breeds included in the study may be considered as a strength. However, the authors could discuss whether breed-specific differences in CRP and NLR levels might exist. There are some publications about breed differences ex. in some Schnauzers. If so, how this could affect the generalizability of the findings? Are there some differences connected with age or sex?

Line 157-163: The results paragraph states that some dogs with structural epilepsy had abnormal basal CRP levels, but the variability within this group is not fully explored. It would be easier to understand if the authors would discuss any potential reasons for this variability, such as differences in the underlying conditions or stages of disease.

Line 196-200: The lack of significant difference in lymphocyte counts between IE subgroups is noted, but the implications of this finding are not fully explored. The authors could discuss whether this suggests that lymphocyte count alone may not be a reliable indicator of epileptic activity and how it might be combined with other markers for obtaining the better diagnostic accuracy.

Line 214-215: The discussion briefly mentions the role of inflammation in epilepsy. The authors could expand on how inflammation might specifically contribute to the pathophysiology of seizures and whether this has been observed in other neurological conditions in dogs. To provide a more comprehensive view for the reader.

Line 214-215:  The authors could further elaborate on the significance of these findings in the broader context of veterinary diagnostics. Specifically, how do these markers compare to other diagnostic tools currently used in everyday veterinary practice?

Line 292-295: Please add limitation paragraphs. The study acknowledges the small sample size as a limitation but does not provide suggestions for how future research could address this issue. In my opinion, the authors could recommend specific strategies, such as multicenter studies or longitudinal designs, to overcome this limitation in future work.

Line 303-304: In conclusion, there is a piece of information that CRP and NLR might be considered potential non-specific markers for cluster seizures and structural epilepsy. Are there any additional factors that might influence these markers and these should be considered in future research?

Line 303-307: CRP and NLR have limited diagnostic value in certain types of seizures, but please support that statement with more detail. For example, adding the information on whether there are specific clinical scenarios where these markers might be more or less useful will be beneficial. Also, suggestion of additional markers that could be evaluated in combination with CRP and NLR may be usefull for practitioners.

Comments on the Quality of English Language

It is ok.

Author Response

Reviewer 2

We want to thank the reviewer for offering us such valuable comment and suggestions! We really appreciate your work! In the revised form we included answers for all the comments we received. We hope that in the actual form we completed all the requests.

Please fin bellow or point by point answers

Comment 1

While the study’s conclusions regarding the use of CRP and NLR as diagnostic tools are supported by the data, the discussion could benefit from further exploration of the long-term implications of using these markers in everyday clinical practice. Thus, it would be beneficial to expand the discussion to include potential long-term implications and limitations of using CRP and NLR as diagnostic tools in canine epilepsy. This could include considerations of how these markers might perform in a broader clinical setting or over extended periods.

Response 1

We appreciate the reviewer’s insightful comment regarding the clinical implications of using CRP as a diagnostic tool. We acknowledge and emphasize that both CRP and NLR are widely accessible and commonly used tests, particularly beneficial for general practitioners when evaluating epileptic patients in the absence of advanced imaging or other definitive diagnostic methods (since this group often lacks access to such advanced tools).

In the modified version of the manuscript, we mentioned between lines 312-323: “Thus, our results suggest that the NLR might be an useful first line tool, especially for the general practitioners in order to preliminary discriminate between the main etiologies of canine epilepsy and therefore adapt the therapeutic approach. Further-more, due the fact that in our dogs both NLR and CRP were significantly increased in cluster patients (p = 0,001) we recommend that dogs presented with a history of isolated seizures and elevated NLR and CRP levels to be managed as cluster seizures patients and be hospitalized for monitoring for at least 48 hours. This approach is crucial because, in veterinary medicine, one of the primary limitations is that owners may not witness or accurately report the true number of seizures their pet has experienced within the last 24 hours. Consequently, these dogs may be at risk of experiencing a neurological emergency such as non-documented cluster seizures.”

Comment 2

The study’s conclusions are based on a relatively small sample size, which may limit the generalizability of the findings. However, future studies with larger sample sizes are recommended to validate the findings and to ensure that the results apply to a broader population of epileptic dogs.

Response 2

Thank you for pointing this out. Despite being performed on a small sample size, our study revealed a couple of statistically significant differences between the groups, which are expected to be more nuanced while being assessed on larger cohorts.

In the limitation section, line 324-335, we modified the manuscript and mentioned that future studies based on larger sample sizes should be performed in order to validate our findings: “ Our study had several limitations. The most important one is represented by the small number of dogs in each group included in this study; further studies based on larger cohorts included in multicentric studies might be useful in highlighting a statistical significance of both the CRP and NLR values according to the etiology of the seizures. Second, the SE group was not subdivided according to the primary pathology, therefore different results might be expected, considering the level of neuroinflammation of various structural pathologies. Third, all the CBCs were performed in the postictal phase, without assessing the values kinetics over the hospitalization days. Similar to human medicine studies, we appreciate that NLR might de-crease and reach a physiological value in the interictal phase [16]. Lastly, we investigate only dogs with self-limiting seizures (IEis and IEcs). Our results are not necessarily applicable in dogs with status epilepticus nevertheless the etiology.”

Comment 3

-          Lack of summary including limitation part

Response 3

We highly appreciate the reviewer’s remarks and we modified the limitation paragraph according to these: Lines 324-335 Our study had several limitations. The most important one is represented by the small number of dogs in each group included in this study; further studies based on larger cohorts included in multicentric studies might be useful in highlighting a statistical significance of both the CRP and NLR values according to the etiology of the seizures. Second, the SE group was not subdivided according to the primary pathology, therefore different results might be expected, considering the level of neuroinflammation of various structural pathologies. Third, all the CBCs were performed in the postictal phase, without assessing the values kinetics over the hospitalization days. Similar to human medicine studies, we appreciate that NLR might de-crease and reach a physiological value in the interictal phase [16]. Lastly, we investigate only dogs with self-limiting seizures (IEis and IEcs). Our results are not necessarily applicable in dogs with status epilepticus nevertheless the etiology.

Comment 4

-       Lack of brief explanation about acute phase protein usage in other species and species-dependent differences in this matter.

Response 4

Thank you for this observation. In the specific comments section, we offer a detailed answer to this remark and highlight the changes made to the text.

Comment 5

Specific Comments:

  1. Line 30-35: In the introduction, there is a mention of the importance of CRP and NLR as potential diagnostic tools. However, it would be helpful if the authors provided a brief comparison with other commonly used inflammatory biomarkers in veterinary medicine, such as fibrinogen or haptoglobin, to contextualize why CRP and NLR were chosen for this study. The Introduction will benefit from that.

Response 5

Thank you for your insightful comments and suggestions regarding our manuscript. We appreciate your feedback, which has helped us refine the discussion of our findings.

We expanded the discussion to include a brief overview of APPs, highlighting their role as non-specific markers of inflammation.

Lines 51-73: “The inflammatory and cytotoxic neural processes contribute to the production of pro-inflammatory cytokines (IL-1β, IL-2, IL-6 and TNF-α) which subsequently lead to increased levels of acute phase proteins (APPs), as part of an early non-specific immune response [6,7]. APPs vary in response magnitude and type across animal species, with the liver being the primary site of their production. In dogs, CRP and serum amyloid A (SAA) are considered as major APPs, while fibrinogen and haptoglobin are classified as moderate APPs. However, only a few studies analyzed the role of different APPs in various neurological diseases. For example, in one study investigating the diagnostic utility of D-dimer and CRP concentration both in blood and cerebrospinal fluid (CSF) in neurological diseases, the authors reported a high CSF D-dimer and CRP concentrations only in dogs with inflammatory conditions. In addition, in intracranial neoplasia the CSF D-dimer was significantly higher when compared with control group [8]. Other diagnostic markers as the genetic ones, SAA, cytokines such as interleukin-17 and CC-motif ligand 19, endocannabinoid receptors and heat shock protein 70 were proposed in some inflammatory neurological conditions [9,10] but they are not currently used in general practice due to their lack of specificity [11]. CRP in particular, exhibits the most significant increase in dogs, making it a highly valuable biomarker for diagnosing and prognosing various diseases in this species [12]. To the best of our knowledge, there are no reports regarding the influence of the individual factors such as age and sex over the CRP concentration in dogs and with the exception of Miniature Schnauzers, which tend to display a slightly higher median CRP concentration [13], interbreed differences are not identified.”

Comment 6

  1. Line 50:Please add some information about acute phase proteins (APPs). Why we are using mostly the major APPs as diagnostic markers? The authors should clearly explain that acute-phase proteins are non-specific markers. In different species of animals, we use different acute-phase proteins because there are species-specific differences. In horses, the main acute-phase protein is serum amyloid A (SAA), the changes in concentration of which are useful in detecting non-specific systemic bacterial infections but are not particularly helpful in fitness monitoring (ex. anabolic index is more helpful in this matter). In cows, SAA is also the main acute-phase protein, as well as in cats. However, in dogs, as in humans, the main acute-phase protein is CRP. Please find accurate literature to built this paragraph.

Response 6

Thank you for this useful suggestion. As mentioned above, in response to your comment, we have added a detailed section on acute-phase proteins (APPs) to the manuscript. Below is a summary of the revisions made:

We expanded the discussion to include a brief overview of APPs, highlighting their role as non-specific markers of inflammation. As you rightly pointed out, different species exhibit variations in their primary acute-phase proteins. The inclusion of this information underscores that CRP, despite being a non-specific marker, is highly relevant and commonly used in veterinary diagnostics, particularly in dogs, where it parallels its use in human medicine. We also emphasized that while CRP is a major APP in dogs, its diagnostic value is particularly pronounced in the context of certain inflammatory conditions, including structural epilepsy. In our revised text, between lines 51-73 , we added: “The inflammatory and cytotoxic neural processes contribute to the production of pro-inflammatory cytokines (IL-1β, IL-2, IL-6 and TNF-α) which subsequently lead to in-creased levels of acute phase proteins (APPs), as part of an early non-specific immune response [6,7].APPs vary in response magnitude and type across animal species, with the liver being the primary site of their production. In dogs, CRP and serum amyloid A (SAA) are considered as major APPs, while fibrinogen and haptoglobin are classified as moderate APPs. However, only a few studies analyzed the role of different APPs in various neurological diseases. For example, in one study investigating the diagnostic utility of D-dimer and CRP concentration both in blood and cerebrospinal fluid (CSF) in neurological diseases, the authors reported a high CSF D-dimer and CRP concentrations only in dogs with inflammatory conditions. In addition, in intracranial neoplasia the CSF D-dimer was significantly higher when compared with control group [8]. Other diagnostic markers as the genetic ones, SAA, cytokines such as interleukin-17 and CC-motif ligand 19, endocannabinoid receptors and heat shock protein 70 were proposed in some inflammatory neurological conditions [9,10] but they are not currently used in general practice due to their lack of specificity [11]. CRP in particular, exhibits the most significant increase in dogs, making it a highly valuable biomarker for diagnosing and prognosing various diseases in this species [12]. To the best of our knowledge, there are no reports regarding the influence of the individual factors such as age and sex over the CRP concentration in dogs and with the exception of Miniature Schnauzers, which tend to display a slightly higher median CRP concentration [13], interbreed differences are not identified.”

Comment 7

  1. Line 57-58: The authors mention that previous studies did not identify a statistical difference in CRP values between IE dogs by frequency of seizures. It would be helpful to discuss why this might be the case and how this finding aligns or conflicts with the current study's results.

Response 7

Thank you for this comment. To the best of our knowledge, there is only one article published by Segers and al. (2017) in which they investigated the difference in CRP concentration of IE dogs by frequency of seizures, which we specified between lines 56-58:” Even though the CRP values were higher in dogs with cluster seizures or SE, the studies have not identified a statistical difference when compared IE dogs by frequency of seizures [8,9].”

 However, two important limitations of this study which could have influenced the lack of a statistical difference are represented by the small sample size (22 dogs with isolated seizures and 16 dogs with cluster seizures) of the groups and the CRP concentrations were measured in different periods of time, not necessary in the first 48 hours after a seizure as in our study.

Comment 8

  1. Line 122-129: please mention that the device was validated for canine species. Including this information about the validation of the assay would strengthen the reliability of the results presented.

Response 8

Thank you for this useful remark. In concordance with the information provided by the manufacturer (Rozanski and Wells- Validation of a point-of-care assay for serum pancreatic lipase and C-reactive protein in the clinical setting), we mentioned that the test is validated for canine species, line 123-124: “A commercially available assay Vcheck Canine CRP (Bionote USA,China) validated for dogs was used in order to measure the CRP concentration [34]. .”

Comment 9

  1. Line 150-155: The diversity of breeds included in the study may be considered as a strength. However, the authors could discuss whether breed-specific differences in CRP and NLR levels might exist. There are some publications about breed differences ex. in some Schnauzers. If so, how this could affect the generalizability of the findings? Are there some differences connected with age or sex?

Response 9

We appreciate the reviewers' thorough evaluation of our manuscript and the insightful comments provided. Indeed, the literature provides one study in which the authors found that healthy miniature schnauzers tend to display slightly higher CRP values in comparison with other breeds. However, other numerous studies cited in our study did not identify a difference in CRP concentration in concordance with breed, age or sex. Moreover, the majority of dogs included in our research was represented by either mixed breed or few individuals from the same breed, therefore we can not draw a conclusion based on these individual factors.

We have revised the manuscript to address this issue, lines 67-73: “CRP in particular, exhibits the most significant increase in dogs, making it a highly valuable biomarker for diagnosing and prognosing various diseases in this species [12]. To the best of our knowledge, there are no reports regarding the influence of the individual factors such as age and sex over the CRP concentration in dogs and with the exception of Miniature Schnauzers, which tend to display a slightly higher median CRP concentration [13], interbreed differences are not identified.”

Comment 10

  1. Line 157-163: The results paragraph states that some dogs with structural epilepsy had abnormal basal CRP levels, but the variability within this group is not fully explored. It would be easier to understand if the authors would discuss any potential reasons for this variability, such as differences in the underlying conditions or stages of disease.

Response 10

Thank you for pointing this out.

We share the reviewer’s concern about this topic and have made the necessary adjustments to improve the manuscript, between lines 237-248: “Similar to the healthy dog group, all dogs with IE had a lower median CRP value in comparison with the SE group one. Furthermore, we found that a half of the dogs diagnosed with either neoplasia or an inflammatory intracranial disease included in the SE group had an abnormal basal blood CRP value, similar to a previous report in which 62% of the dogs with structural epilepsy had a higher blood CRP concentration [15]. Cavalerie R. et al. (2024) showed that CRP has a limited diagnostic value in dogs with meningoencephalitis of unknown origin, where only 30% of dogs had an increase of the blood CRP concentration [39]. However, our aim was not to subdivide the SE group by pathologies, but to compare it with the findings from other epileptic groups. Hence, the high CRP concentration identified in our study might be a consequence of the heterogeneity of the SE group (e.g. differences in the underlying conditions or stages of disease).”

Comment 11

  1. Line 196-200: The lack of significant difference in lymphocyte counts between IE subgroups is noted, but the implications of this finding are not fully explored. The authors could discuss whether this suggests that lymphocyte count alone may not be a reliable indicator of epileptic activity and how it might be combined with other markers for obtaining the better diagnostic accuracy.

Response 11

We are grateful for the reviewer’s suggestion regarding the implication of analyzing lymphocyte count alone.

In the revised form of the manuscript, we clarified that between lines 279-287: “In our study, all epileptic groups, regardless of the etiology, had an increased NLR (p = 0.005). The analysis of NLR is preferred over lymphocyte or neutrophil count alone because it offers a more nuanced and comprehensive assessment of the body's immune response [17] that might be helpful in diagnosis, prognosis, and monitoring of the disease. Indeed, our results showed no significant differences on the lymphocyte count when both groups and subgroups were compared. Therefore, the increased NLR was based only on the neutrophil count. Thus, we suggest that the lymphocyte count may not be a reliable indicator of epileptic activity and should be combined with other markers for obtaining the better diagnostic accuracy.”

Comment 12

  1. Line 214-215: The discussion briefly mentions the role of inflammation in epilepsy. The authors could expand on how inflammation might specifically contribute to the pathophysiology of seizures and whether this has been observed in other neurological conditions in dogs. To provide a more comprehensive view for the reader.

Response 12

We highly appreciate your recommendation empathizing the role of inflammation in epilepsy. In the revised form of the manuscript we mentioned how the inflammation might specifically contribute to the pathophysiology of seizures between lines 273-278 : “Previous researches showed that the infiltration of inflammatory cells into the hippocampus, the breakdown of the blood-brain barrier (BBB) and the increasing levels of leukocytes predispose to neurodegeneration [47–49]. Furthermore, the potential of therapies such as anti-inflammatory drugs is investigated in order to establish if it could modulate the upregulation of the BBB by COX-2 inhibition [50,51].” In addition to lines 281-289: “As it was shown in previous reports, immunosuppression and remodulation generate a tumor microenvironment which will lead to neuroinflammation [52–54]. Furthermore, it has been suggested that neutrophils are involved in the early stages of the pathogenesis, accelerating the infiltration of the inflammatory cells and therefore generating the lesions [55]. Thus far, the contribution of the neutrophils in the neuronal regulation of the autoimmunity remains unclear [56,57]. As a result of the similarities of the inflammatory brain diseases from humans and dogs [43], we presume that the NLR values were related to the neuronal inflammation.”.

Moreover, we updated the lines 59-69: “For example, in one study investigating the diagnostic utility of D-dimer and CRP concentration both in blood and cerebrospinal fluid (CSF) in neurological diseases, the authors reported a high CSF D-dimer and CRP concentrations only in dogs with inflammatory conditions. In addition, in intracranial neoplasia the CSF D-dimer was significantly higher when compared with control group [8]. Other diagnostic markers as the genetic ones, SAA, cytokines such as interleukin-17 and CC-motif ligand 19, endocannabinoid receptors and heat shock protein 70 were proposed in some inflammatory neurological conditions [9,10] but they are not currently used in general practice due to their lack of specificity [11]. CRP in particular, exhibits the most significant increase in dogs, making it a highly valuable biomarker for diagnosing and prognosing various diseases in this species [12].”.

Comment 13

  1. Line 214-215:  The authors could further elaborate on the significance of these findings in the broader context of veterinary diagnostics. Specifically, how do these markers compare to other diagnostic tools currently used in everyday veterinary practice?

Response 13

Thank you for highlighting the importance of these markers in the context of veterinary diagnostics.

We have taken this feedback into account and made the following changes to enhance the clarity and accuracy of our work, between lines 313-324: “Thus, our results suggest that the NLR might be an useful first line tool, especially for the general practitioners in order to preliminary discriminate between the main etiologies of canine epilepsy and therefore adapt the therapeutic approach. Further-more, due the fact that in our dogs both NLR and CRP were significantly increased in cluster patients (p = 0,001) we recommend that dogs presented with a history of isolated seizures and elevated NLR and CRP levels to be managed as cluster seizures patients and be hospitalized for monitoring for at least 48 hours. This approach is crucial because, in veterinary medicine, one of the primary limitations is that owners may not witness or accurately report the true number of seizures their pet has experienced within the last 24 hours. Consequently, these dogs may be at risk of experiencing a neurological emergency such as non-documented cluster seizures.”

Comment 14

  1. Line 292-295: Please add limitation paragraphs. The study acknowledges the small sample size as a limitation but does not provide suggestions for how future research could address this issue. In my opinion, the authors could recommend specific strategies, such as multicenter studies or longitudinal designs, to overcome this limitation in future work.

Response 14

Thank you for this useful remark.

We updated with the reviewer’s recommendation, between lines 325-336: “Our study has a limitation paragraph based on four important limitations, including the sample size, the lack of subdividing the structural epilepsy group, the lack of assessing the values kinetics over the hospitalization days and the exclusion of dogs presented in status epilepticus. Our study had several limitations. The most important one is represented by the small number of dogs in each group included in this study; further studies based on larger cohorts included in multicentric studies might be useful in highlighting a statistical significance of both the CRP and NLR values according to the etiology of the seizures. Second, the SE group was not subdivided according to the primary pathology, therefore different results might be expected, considering the level of neuroinflammation of various structural pathologies. Third, all the CBCs were performed in the postictal phase, without assessing the values kinetics over the hospitalization days. Similar to human medicine studies, we appreciate that NLR might de-crease and reach a physiological value in the interictal phase [16]. Lastly, we investi-gate only dogs with self-limiting seizures (IEis and IEcs). Our results are not necessari-ly applicable in dogs with status epilepticus nevertheless the etiology.”

Comment 15

  1. Line 303-304: In conclusion, there is a piece of information that CRP and NLR might be considered potential non-specific markers for cluster seizures and structural epilepsy. Are there any additional factors that might influence these markers and these should be considered in future research?

Response 15

We appreciate the reviewer’s attention to additional factors that might influence CRP or NLR values in the process of establishing a diagnosis of an epileptic patient. However, all epileptic dogs fulfilled at least the first tier according to IVETF consensus, as we specified in materials and methods section between lines 103-109:  “The included dogs had a history of at least two unprovoked epileptic seizures occurring at least 24 hours apart, complete physical and neurological examination, complete blood cell count, serum biochemical profile, urinalysis, with supplementary thoracic radiography, abdominal ultrasound and non-invasive blood pressure measurement; therefore all dogs completed the first confidence level according to International Veterinary Epilepsy Task Force (IVETF) consensus [26].” As a consequence, we excluded the possibility that another potential factors influenced the values of these non-specific markers for both the IE group presented with cluster seizures or the structural epilepsy cohort.

We have revised the manuscript to address this issue more clearly, lines 338-344: “In conclusion, in the absence of other concurrent inflammatory diseases, blood CRP concentration and NLR value might be considered as potential non-specific markers and used in the diagnostic approach of dogs with structural epilepsy or cluster seizures in dogs with idiopathic epilepsy. Dogs diagnosed with IEis and high CRP and NLR may be subject to non-documented cluster seizures. Both measurements have limited diagnostic value in dogs experiencing reactive seizures.”

Comment 16

  1. Line 303-307: CRP and NLR have limited diagnostic value in certain types of seizures, but please support that statement with more detail. For example, adding the information on whether there are specific clinical scenarios where these markers might be more or less useful will be beneficial. Also, suggestion of additional markers that could be evaluated in combination with CRP and NLR may be usefull for practitioners.

Response 16

We fully agree with the reviewer’s observation regarding adding information on the specific clinical scenarios where these markers might be more useful.

As a consequence, we have incorporated the suggested changes into the manuscript between lines 316-324: “Furthermore, due the fact that in our dogs both NLR and CRP were significantly in-creased in cluster patients (p = 0,001) we recommend that dogs presented with a history of isolated seizures and elevated NLR and CRP levels to be managed as cluster seizures patients and be hospitalized for monitoring for at least 48 hours. This approach is crucial because, in veterinary medicine, one of the primary limitations is that owners may not witness or accurately report the true number of seizures their pet has experienced within the last 24 hours. Consequently, these dogs may be at risk of experiencing a neurological emergency such as non-documented cluster seizures.”

And lines 338-344: “In conclusion, in the absence of other concurrent inflammatory diseases, blood CRP concentration and NLR value might be considered as potential non-specific markers and used in the diagnostic approach of dogs with structural epilepsy or clus-ter seizures in dogs with idiopathic epilepsy. Dogs diagnosed with IEis and high CRP and NLR may be subject to non-documented cluster seizures. Both measurements have limited diagnostic value in dogs experiencing reactive seizures.”

We hope the revised form of the manuscript will satisfy all the queries. Once more, we thank you for your work in improving the quality and depth of the study.

The authors

Round 2

Reviewer 1 Report

Comments and Suggestions for Authors

I thank the authors for providing the clarifications I needed. The changes made to the title and the explanations included in the discussion and conclusions have clarified the objectives and results of the work. I found Table 1 very interesting as it helps clarify the results and provides insights for further research.